# Anti-Inflammatory Effects of Idebenone Attenuate LPS-Induced Systemic Inflammatory Diseases by Suppressing NF-κB Activation

**DOI:** 10.3390/antiox13020151

**Published:** 2024-01-25

**Authors:** Yumin Choi, Young-Lai Cho, Sujeong Park, Minkyung Park, Keun-Seok Hong, Young Jun Park, In-Ah Lee, Su Wol Chung, Heedoo Lee, Seon-Jin Lee

**Affiliations:** 1Environmental Disease Research Center, Korea Research Institute of Bioscience and Biotechnology (KRIBB), Daejeon 34141, Republic of Korea; u_min98@naver.com (Y.C.); ylcho@kribb.re.kr (Y.-L.C.); sjpark01@kribb.re.kr (S.P.); mingk@kribb.re.kr (M.P.); hongs06@naver.com (K.-S.H.); pyj71@kribb.re.kr (Y.J.P.); 2Department of Functional Genomics, University of Science and Technology (UST), Daejeon 34113, Republic of Korea; 3Department of Chemistry, Kunsan National University, Gunsan 54150, Republic of Korea; leeinah@kunsan.ac.kr; 4Department of Biological Sciences, College of Natural Sciences, University of Ulsan, Ulsan 44610, Republic of Korea; swchung@ulsan.ac.kr; 5Department of Biology and Chemistry, Changwon National University, Changwon 51140, Republic of Korea; leehd@changwon.ac.kr

**Keywords:** anti-inflammation, inflammatory disease model, NF-κB, ROS, idebenone

## Abstract

Inflammation is a natural protective process through which the immune system responds to injury, infection, or irritation. However, hyperinflammation or long-term inflammatory responses can cause various inflammatory diseases. Although idebenone was initially developed for the treatment of cognitive impairment and dementia, it is currently used to treat various diseases. However, its anti-inflammatory effects and regulatory functions in inflammatory diseases are yet to be elucidated. Therefore, this study aimed to investigate the anti-inflammatory effects of idebenone in cecal ligation puncture-induced sepsis and lipopolysaccharide-induced systemic inflammation. Murine models of cecal ligation puncture-induced sepsis and lipopolysaccharide-induced systemic inflammation were generated, followed by treatment with various concentrations of idebenone. Additionally, lipopolysaccharide-stimulated macrophages were treated with idebenone to elucidate its anti-inflammatory effects at the cellular level. Idebenone treatment significantly improved survival rate, protected against tissue damage, and decreased the expression of inflammatory enzymes and cytokines in mice models of sepsis and systemic inflammation. Additionally, idebenone treatment suppressed inflammatory responses in macrophages, inhibited the NF-κB signaling pathway, reduced reactive oxygen species and lipid peroxidation, and normalized the activities of antioxidant enzyme. Idebenone possesses potential therapeutic application as a novel anti-inflammatory agent in systemic inflammatory diseases and sepsis.

## 1. Introduction

Inflammation is a natural immune response to injury, infection, or irritation [1,2]. However, persistent inflammation can lead to organ dysfunction [2,3], contributing to several major diseases [4], indicating the need for the development of effective treatments for inflammation.

Sepsis is defined as a life-threatening organ dysfunction caused by the dysregulation of the host response to infection, resulting in organ dysfunction and failure [5,6]. Upon recognizing an infection, immune cells such as macrophages and dendritic cells express pattern recognition receptors (PRRs) to initiate host defenses. Pathogen and damage-associated molecular patterns (PAMPs and DAMPs, respectively) interact with Toll-like receptors (TLRs) in order to induce inflammation and cytokine storms [5,6,7]. NF-κB activation by these interactions leads to excessive cytokine production, including tumor necrosis factor-α (TNF-α), interleukin-1 beta (IL-1β), interleukin-6 (IL-6), prostaglandin E2 (PGE2), and nitric oxide (NOx) [5,6,7]. Inflammatory cytokines induce the production of reactive oxygen species (ROS), damaging cells, DNA, and tissues [8,9]. Additionally, ROS activate signaling pathways, thereby inducing inflammatory responses and tissue damage [10,11]. However, hyperinflammatory responses can be treated using anti-inflammatory drugs; therefore, the discovery of novel anti-inflammatory drugs may provide an effective treatment strategy for inflammatory diseases.

In the early 1980s, idebenone was developed to treat cognitive decline and dementia [12]. Coenzyme Q10 (CoQ10), which displays antioxidant activity, faces solubility and absorption issues [13,14]. Therefore, idebenone was developed to address the solubility issue of CoQ10 and enhance mitochondrial electron transport, ATP synthesis, and energy metabolism, improve ROS scavenging, and reduce oxidative stress. Although idebenone shows therapeutic promise for neuro-inflammation and other conditions [15,16], its effects in systemic inflammatory diseases are yet to be elucidated. Idebenone transfers electrons within the mitochondrial respiratory chain [17] and serves as a fat-soluble antioxidant, scavenging ROS and influencing cell metabolism [18]. Previous studies have found that CoQ10 regulates the expression of IL-1β and TNF-α [19]. Given that idebenone is similar to CoQ10, we hypothesized that idebenone may also modulate the expression of inflammatory cytokines. Idebenone is globally utilized in clinical trials and treatments [20], and has demonstrated therapeutic potential in various conditions, such as LHON, dementia, skin aging, and Friedrich’s ataxia [21,22,23].

Although idebenone has shown therapeutic effects in neuroinflammation and other diseases, its effect in systemic inflammatory diseases is yet to be elucidated. Therefore, this study aimed to investigate the anti-inflammatory effects of idebenone in murine models of cecal ligation puncture (CLP)-induced sepsis, lipopolysaccharide (LPS)-induced systemic inflammation, and LPS-induced macrophage cell lines.

## 2. Materials and Methods

### 2.1. Reagents and Antibodies

Idebenone (15475), mito-TEMPO (16621), and diphenyleneiodonium chloride (DPI; 81050) were purchased from Cayman Chemical Company (Ann Arbor, MI, USA). LPS (L2630) was purchased from Sigma-Aldrich (St. Louis, MO, USA), and *N*-acetyl-L-cysteine (NAC; A0905) was purchased from Tokyo Chemical Industry Co. (Tokyo, Japan). The following antibodies were used for Western blotting and immunohistochemical analyses: β-actin (sc-47778), SOD1 (sc-11407), SOD2 (sc-30080), Catalase (sc-50508), and α-tubulin (sc-8035) (Santa Cruz Biotechnology Inc., Santa Cruz, CA, USA); iNOS (13120s), COX-2 (12282s), TNF-α (3707s), IL-1β (12242s), IκBα (4812s), p-IκBα (2859s), IKKβ (2370s), p-IKKβ (2078s), p65 (8242s), p38 (8690s), p-p38 (4511s), JNK (9252s), p-JNK (4668s), ERK (4695s), p-ERK (4370p), AKT (4691s), and p-AKT (4060s) (Cell Signaling Technology Co., Danvers, MA, USA); and Lamin A (ab26300) (Abcam Ltd., Cambridge, UK). Permount was purchased from Fisher Scientific (Morris Plains, NJ, USA).

### 2.2. Animal Experimentations

All animal handling and experimentations were in accordance with ethical guidelines and approved by the Animal Care and Use Institutional Review Committee of Korea Research Institute of Bioscience and Biotechnology (approval number: KRIBB-AEC-22254). Eight-week-old female C57BL/6 mice were purchased from Orient (Seongnam-si, Gyeonggi-do, Republic of Korea) and housed in plastic cages in a room with controlled temperatures (22 1 ± 2 °C) and humidity (55 ± 5%), and also maintained on a reverse 12 h light/dark cycle. For the LPS induction of inflammation, systemic inflammation was induced by intraperitoneal injection of lipopolysaccharide (LPS) at a dose of 10 mg/kg in a volume of 100 μL sterile saline in the experimental groups. To obtain 200 µg of idebenone from the 40 mM idebenone stock solution in DMSO, approximately 0.6 µL of the idebenone stock solution is needed. This was then diluted with PBS to a total volume of 1 mL in order to overcome the insolubility of idebenone and also for fluid resuscitation. Idebenone (dissolved in DMSO, 10 mg/kg) was administered intraperitoneally to the treatment group immediately following LPS injection. A control group received intraperitoneal injections of the vehicle. After the injection, mice were monitored for 12 days post-treatment, and blood, tissues, or relevant samples were collected for subsequent analyses. For Cecal ligation and puncture (CLP) induction of sepsis, CLP was surgically performed on mice according to the original protocol as developed by Lee et al. [24], with some modifications. Briefly, the mice were anesthetized using an intraperitoneal injection of avertin (500 mg/kg). A midline incision was made, and the cecum was ligated 1 cm from the apex and punctured (one hole) using a 23 g needle. Thereafter, a small fecal mass from the punctured cecum was gently squeezed out to ensure patency. The cecum was relocated, and 6–0 sutures were used to close the peritoneum and skin. Mice in the sham group underwent incisions and cecal exteriorizations only. Around 24 h after the sham and CLP operations, the mice were either injected with 1 mL of idebenone (10 mg/kg) that was prepared in a phosphate-buffered saline (PBS) for anti-inflammatory drug effects and fluid resuscitation, or with 1 mL of PBS, only subcutaneously for fluid resuscitation. The survival rate was assessed every 12 h over 5 days (CLP) and 12 days (LPS). Pre- and post-operatively, all mice had unlimited access to food and water.

### 2.3. Hematoxylin and Eosin (HE) Staining

Mouse tissue in the lung, liver, kidney, and colon was cut into 7 µm thick sections, followed by histochemical staining using HE. Tissue sections were deparaffinized using graded concentrations of xylene (100, 95, and 70%) for 5 min each. For nuclear staining, the sections were stained with hematoxylin solution for 5 min and washed with flowing water. Hematoxylin was removed using 1% HCl and 1% ammonia solutions, followed by immersion in an eosin solution for 2 min in order to stain the cytoplasm. Thereafter, the sections were dehydrated using graded concentrations of alcohol (70, 95, and 100%), followed by treatment with xylene which was performed three times (3 min each) to remove any alcohol. Finally, the sections were mounted with glass coverslips and viewed using an Olympus BX43 microscope (Olympus, Tokyo, Japan). Photographs were taken with an Olympus DP72 camera (Olympus).

### 2.4. Immunohistochemistry for iNOS

Immunohistochemical staining was performed on 4–6 μm thick sections that were fixed in 10% formalin and embedded in paraffin. Briefly, the sections were deparaffinized by washing with xylene three times (5 min each), followed by treatment with graded concentrations of alcohol (100, 95, and 70%) for 10 min each. After rinsing twice with distilled water (dH_2_O) for 5 min, the sections were treated with 3% hydrogen peroxide for 10 min in order to block peroxidase activity. After washing twice with dH_2_O for 5 min each, the sections were blocked with 200 µL of blocking solution with 1.5% bovine serum albumin (BSA) in Tris-buffered saline (TBS; 50 mM Tris and 150 mM NaCl, pH 7.6) containing 0.1% Tween 20 (TBS-T) for 1 h at room temperature (25 °C), followed by incubation with 200 µL of the primary antibody against iNOS overnight at 4 °C. Thereafter, the samples were washed three times with TBS-T for 5 min, followed by incubation with 200 µL of a secondary antibody for 30 min at room temperature. After washing three times with TBS-T for 5 min, the samples were treated with the ABC reagent kit (Mouse IgG PK-6102, Vector Laboratories, Burlingame, CA, USA). Thereafter, the sections were washed three times with TBS-T and stained with 100 µL of DAB, rinsed twice with dH_2_O for 5 min, treated twice with 95% ethanol for 10 s, and then twice again with 100% ethanol. Finally, the samples were treated with xylene twice for 10 s each, mounted with coverslips, and viewed using a confocal laser scanning microscope (Carl Zeiss, Jena, Germany), the LSM 510 META module, and 20× lenses. Microscope images were processed using the Start LSM Image Browser (version 4.2.0.121).

### 2.5. Cells and Cell Culture

The murine macrophage cell lines RAW 264.7 and J774A.1 were obtained from the Korean Cell Line Bank (Seoul, Republic of Korea). RAW 264.7 cells were cultured in Dulbecco’s modified Eagle’s medium (SH30243.01, Hyclone, Logan, UT, USA), supplemented with 10% non-heat inactivated calf serum (26170-043, Gibco™, Waltham, MA, USA) and 1% antibiotics (LS 203-01, WELGENE Inc., Gyeongsan, Republic of Korea). The J774A.1 cell line was cultured in Dulbecco’s modified Eagle’s medium (SH30243.01, Hyclone, Logan, UT, USA), supplemented with 10% non-heat inactivated fetal bovine serum (SH30919.03, Hyclone, Logan, UT, USA), 1% antibiotics (LS 203-01, WELGENE Inc., Gyeongsan, Republic of Korea), and 25 mM HEPES (15630-080, Gibco™, Waltham, MA, USA) at 37 °C in a humidified incubator with 5% CO_2_. The cells were treated with LPS at 1 μg/mL.

### 2.6. Assessment of NO Metabolite Levels

The levels of nitrite, a stable oxidized product of NO, were measured in the culture media using the Griess reagent. Both serum nitrite and nitrate (NOx) concentration were determined with a nitrate reductase-based colorimetric assay kit (Alexis, Los Angeles, CA, USA). Triplicates of each sample were incubated with the same volume of sulfanilamide and *N*-(1-Naphthyl) ethylenediamine solution. After 5–10 min, absorbance levels were measured at 550 nm using a SpectraMax ABS Plus microplate reader (Molecular Devices, San Jose, CA, USA).

### 2.7. Western Blot Analysis

Total cell lysates and tissue were lysed on ice using a radioimmunoprecipitation assay (RIPA) lysis buffer (50 mM Tris-HCl, pH 7.6, 150 mM NaCl, 10% glycerol, 1% Nonidet P-40, 5 mM EDTA, 1 mM DTT, 100 mM NaF, 2 mM sodium pyrophosphate, 20 mM β-glycerophosphate, 2 mM sodium orthovanadate) and 1× protease inhibitor cocktail (Sigma-Aldrich) for 30 min. The protein contents of cell and tissue lysates were quantified using the Pierce BCA Protein Assay kit (23209, Thermo Scientific, Chicago, IL, USA). Thereafter, proteins (10–50 μg) were separated on a 6–15% gel via sodium dodecyl sulfate-polyacrylamide gel electrophoresis and transferred to a polyvinylidene difluoride membrane using a Trans-Blot^®^ TurboTM Transfer pack (Bio-Rad, Hercules, CA, USA). The membranes were blocked with 5% skim milk/TBST for 1 h and incubated with primary antibodies overnight at 4 °C. After three washes, membranes were incubated with horseradish peroxidase (HRP)-conjugated secondary antibodies for 40 min at RT. After washing for 2 h, the protein bands were visualized using Clarity Western ECL Substrate (1705061; Bio-Rad, Hercules, CA, USA), and chemiluminescent images were captured by Fusion Solo S (Vilber Lourmat, Paris, France), following the instruction manual.

### 2.8. Preparation of Cytoplasmic and Nuclear Fractions

After treating RAW264.7 cells with lipopolysaccharide (LPS, 1 μg/mL) in the presence or absence of idebenone (20 μM) for 30 min, cellular nuclear and cytoplasmic fractions were isolated using a commercial kit (Active Motif, Carlsbad, CA, USA), following the manufacturer’s instructions. The fractions were obtained to study the translocation of NF-κB, a crucial transcription factor in the inflammatory response. To ensure consistent protein loading for subsequent Western blot analysis, the concentration of each fraction was determined using a Pierce BCA assay (Thermo Scientific). Fractionation efficiency was assessed by Western blotting using α-tubulin as a loading control for the cytoplasmic fraction and Lamin A as a loading control for the nuclear fraction.

### 2.9. Determination of the Concentrations of TNF-α, IL-1β, IL-6, and PGE_2_

The concentrations of TNF-α, IL-1β, and IL-6 in the culture medium were determined using the Duoset ELISA system (DY410, DY406, and DY401, R&D Systems, Minneapolis, MN, USA). PGE_2_ concentration was determined using the PGE_2_ ELISA kit (514010, Cayman Chemicals, Ann Arbor, MI, USA), according to the manufacturer’s instructions. The absorbance was measured at 450 nm using a SpectraMax ABS Plus microplate reader (Molecular Devices, San Jose, CA, USA).

### 2.10. Immunofluorescence Assay

RAW 264.7 cells were grown on 12 mm cover glasses in 12-well culture plates containing DMEM, followed by pretreatment with idebenone (20 μM) for 2 h and treatment with LPS (1 μg/mL). After 1 h, the cells were washed twice with phosphate-buffered saline (PBS), fixed with 4% paraformaldehyde for 10 min, and permeabilized with 0.2% Triton X-100 in PBS for 20 min. Thereafter, the cells were blocked with 3% BSA for 1 h, followed by incubation with the appropriate primary antibody overnight at 4 °C. After washing, the cells were incubated with an Alexa Fluor 594-conjugated secondary antibody (A-21442, Thermo Fisher Scientific, Waltham, MA, USA) for 30 min. After washing twice with PBS, the cells were mounted with a Vectashield mounting medium containing DAPI (H-1500, Vector Laboratories, Newark, CA, USA), and images were acquired using a fluorescence microscope (Olympus, BX53, New York, NY, USA).

### 2.11. Measurement of ROS Production

ROS production by RAW 264.7 cells was detected using a DCFDA/H_2_DCFDA-Cellular ROS Assay kit (ab113851, Abcam, Cambridge, UK), according to the manufacturer’s instructions. Briefly, cells were seeded in six-well plates (35 mm plates) at a density of 1 × 10^6^ cells/well, followed by treatment with LPS (1 μg/mL), idebenone (5, 10, 20, and 40 μM), *N*-Acetyl-L-cysteine (NAC) (5 mM), mito-TEMPO (50 μM), and Diphenyleneiodonium chloride (DPI) (10 μM). The cells were harvested using trypsin and washed with cold PBS. Finally, the cells were stained with 20 μM of 2′,7′-dichlorofluorescin diacetate (DCFDA) for 20 min and then analyzed using a NovoCyte Flow Cytometer (150014, Agilent Technologies Inc., Santa Clara, CA, USA).

### 2.12. Lipid Peroxidation Assay

The malondialdehyde (MDA) content of the cell lysates was assessed using a lipid peroxidation assay kit (ab83366, Abcam, CA, UK), according to the manufacturer’s instructions.

### 2.13. Statistical Analysis

Quantitative data are presented as the mean ± standard deviation (SD). Quantification of the Western blots were analyzed using ImageJ software (V1.8.0), and the level of the indicated protein was normalized to β-actin. Significant differences between groups were determined using two-tailed unpaired Student’s *t*-test. CLP-induced and LPS-induced mouse serum and tissue sample analyses were carried out by one-way ANOVA with Tukey’s Method, using the GraphPad Prism 9 statistical program. Statistical significance was set at * *p* < 0.05, ** *p* < 0.01, and # *p* < 0.001.

## 3. Results

### 3.1. Idebenone Reduces Inflammatory Disease-Related Mortality and Protects against Tissue Damage In Vivo

In this study, we assessed the therapeutic effects of idebenone in mice with CLP-induced sepsis and LPS-induced acute inflammation. After an intraperitoneal injection of idebenone (10 mg/kg), the group, compared to the PBS group, following CLP surgery, showed an improvement in survival rate of approximately 40–50% by the third day (Figure 1A). Similarly, the LPS-only group exhibited an approximately 80% survival rate on the second day after LPS injection, which decreased to approximately 50% on the fourth day and was maintained at 30% on the sixth day. In contrast, the LPS plus idebenone group consistently maintained an 80% survival rate starting from the fourth day after LPS injection (Figure 1B). Collectively, these results indicate the therapeutic potential of idebenone against inflammatory diseases in both models. Additionally, we examined the effects of idebenone on tissue damage using the same models. Immunohistochemistry revealed reduced iNOS staining in lung, liver, kidney, and colon tissues in the idebenone-treated CLP group (Figure 1C). Moreover, HE staining showed that idebenone-treated mice exhibited suppressed inflammatory reactions, in contrast to the disrupted lung and colon tissues in the LPS group (Figure 1D). Overall, these results suggest that idebenone may be effective in protecting against tissue damage that is caused by inflammatory diseases.

### 3.2. Idebenone Inhibits Inflammatory Responses in Mouse Models of CLP- and LPS-Induced Inflammation

To examine the anti-inflammatory effects of idebenone on CLP- and LPS-induced inflammatory responses, we examined the serum and tissue levels of NOx, the inflammatory enzyme cyclooxygenase-2 (COX-2), and the inflammatory cytokine IL-1β. Idebenone treatment demonstrated inhibition of NOx levels in the serum of the CLP-induced mouse model (Figure 2A), as well as a significant suppressive effect on the upregulation of inflammatory cytokines, including COX-2 and IL-1beta, in lung, liver, and kidney tissues (Figure 2B–D). Moreover, idebenone treatment effectively attenuated the LPS-induced inflammatory response, as indicated by reduced serum NOx levels (Figure 2E), and suppressed expression of inflammatory cytokines in lung, liver, and kidney tissues (Figure 2F–H). Collectively, these results indicate that idebenone enhances survival in mice with CLP- and LPS-induced systemic inflammatory diseases by suppressing inflammatory responses. To confirm the anti-inflammatory effects at the protein level, we examined the serum levels of inflammatory cytokines using ELISA. Expectedly, idebenone treatment significantly suppressed both CLP- and LPS-induced increases in the serum levels of inflammatory cytokines, including TNF-α, IL-6, IL-1β, and PGE_2_, compared with that in the PBS-treated CLP and LPS groups (Figure 3A–H). Overall, these results indicate that idebenone can suppress the secretion of inflammatory cytokines in both CLP-induced sepsis and LPS-induced systemic inflammatory disease.

### 3.3. Idebenone Inhibits LPS-Induced Expression of Proinflammatory Mediators in Macrophages

Macrophages play a crucial role in driving inflammatory diseases by generating NOx, prostaglandin mediators, and inflammatory cytokines during inflammatory responses [4,6]. To validate the in vivo results, we examined the anti-inflammatory effect of various concentrations of idebenone in LPS-stimulated macrophage cell lines, RAW 264.7 and J774A.1. After 18 h, NOx production in the cell culture medium was measured. Notably, idebenone treatment significantly suppressed LPS-induced NOx production by macrophages in a dose-dependent manner (Figure 4A). Additionally, Western blot analysis showed that idebenone treatment attenuated the LPS-induced expression of iNOS, COX-2, TNF-α, and IL-1β in a concentration-dependent manner (Figure 4B). Moreover, ELISA showed that idebenone treatment significantly inhibited an LPS-induced increase in the secretion of inflammatory cytokines (TNF-α, IL-6, IL-1β, and PGE_2_) in the cell culture medium in a dose-dependent manner (Figure 4C–F). Consistent with the in vivo results, idebenone effectively reduced the levels of inflammatory molecules and cytokines in vitro. Overall, the ability of idebenone to inhibit the production of inflammatory mediators and cytokines underscores its potential to mitigate systemic inflammatory diseases. Moreover, these results were confirmed in J774A.1 cells (Appendix A).

### 3.4. Idebenone Inhibites LPS-Induced NF-κB Activation in Macrophages

A decrease in the expression of inflammatory factors is closely linked to the LPS signaling pathway. NF-κB is a pivotal transcription factor that regulates the expression of various pro-inflammatory genes in response to LPS [5,6,7]. Therefore, we examined the effect of idebenone on the LPS signaling pathway, with a focus on the NF-κB signal. Specifically, we investigated the effects of idebenone on the nuclear factor of kappa light polypeptide gene enhancers in B-cell inhibitors and alpha (IκBα) phosphorylation and degradation. Idebenone treatment (20 µM) significantly inhibited IκBα phosphorylation and degradation in LPS-stimulated macrophages within approximately 10–15 min (Figure 5A). Additionally, we examined the phosphorylation of inhibitors of the nuclear factor kappa-B kinase subunit beta (IKKβ), an upstream signal of IκBα, and found that idebenone treatment significantly suppressed IKKβ phosphorylation (Figure 5B). Moreover, we assessed the effect of idebenone on the nuclear translocation of the NF-κB p65 subunit. Western blotting and confocal microscopy showed that idebenone suppressed the translocation of cytosolic p65 to the nucleus, a downstream signal of IκBα (Figure 5C,D). However, idebenone did not significantly affect other LPS-related signals, including protein kinase B (AKT), extracellular signal-regulated kinase (ERK), c-Jun N-terminal kinase (JNK), and p38 mitogen-activated protein kinase (p38) (Figure 5E). Collectively, these results indicate that idebenone modulates the NF-κB signaling pathway in order to suppress LPS-induced inflammatory responses.

### 3.5. Idebenone Suppresses ROS Generation and Lipid Peroxidation in LPS-Stimulated RAW264.7 Cells

LPS treatment increased ROS and lipid peroxide production, contributing to oxidative stress through an imbalanced redox state. Idebenone is known for its ROS-scavenging effects [25]. To assess the antioxidant activity of idebenone, we examined ROS generation and lipid peroxidation in LPS-stimulated macrophages. Fluorescence-activated cell sorting (FACs) showed that idebenone treatment reduced ROS levels in LPS-treated cells in a time- and dose-dependent manner (Figure 6A,B). Moreover, the ROS scavenging activity of idebenone was comparable to those of NAC, mito-TEMPO, and DPI (Figure 6C). Malonaldehyde (MDA), an end product of lipid peroxidation, serves as an index of lipid peroxidation. Idebenone treatment caused a concentration-dependent decrease in MDA levels in LPS-stimulated macrophages (Figure 6D). ROS are connected to antioxidant enzymes, such as SOD, which convert superoxide anions to H_2_O_2_. Western blotting revealed that idebenone treatment reinstates the expression of antioxidant enzymes, including SOD1, SOD2, and catalase, in LPS-stimulated macrophages, effectively restoring their levels to normal (Figure 6E). Collectively, these results showed that idebenone effectively inhibits ROS generation and lipid peroxidation by modulating antioxidant enzyme activity in LPS-stimulated macrophages.

## 4. Discussion

Idebenone, a synthetic derivative of CoQ10, has received considerable attention owing to its potential use in various diseases, including neurodegenerative disorders. However, recent findings suggest that idebenone exerts anti-inflammatory effects [15,25]. Therefore, we conducted in vitro and in vivo experiments to investigate the anti-inflammatory effects of idebenone.

CoQ10 treatment has been shown to reduce joint inflammation and inflammatory cytokine levels and improve histopathological outcomes in arthritis models [26]. Similarly, idebenone treatment showed protective effects in an acute lung injury model by suppressing both inflammation and oxidative stress [27]. Additionally, idebenone treatment reduced proinflammatory cytokine levels and ER stress in the colon in a murine model of chronic colitis. Consistent with previous findings [28], idebenone treatment improved tissue conditions and decreased the levels of inflammatory cytokines and NOx in mice models of CLP-induced sepsis and LPS-induced inflammation in the present study. Collectively, these findings indicate that idebenone has therapeutic efficacy in inflammatory models.

The anti-inflammatory effects of idebenone appear to be associated with its multifaceted mechanisms of action. For instance, idebenone exerts antioxidant effects to relieve oxidative stress. This is a characteristic of inflammatory reactions. Similarly, idebenone exerted significant ROS scavenging activity in the present study, which was comparable to those of several ROS inhibitors. The antioxidants SOD and catalase are responsible for the direct removal of active superoxide radicals to protect against lipid peroxidation [29,30]. Consistent with previous findings [31,32], idebenone treatment restored LPS-induced increases in the expression of antioxidant enzymes to normal levels in the present study.

The ability of idebenone to scavenge ROS and reduce oxidative stress may influence the activation of the NF-κB pathway. During ROS scavenging, idebenone can potentially inhibit upstream events that trigger NF-κB activation, thus preventing the subsequent transcription of proinflammatory genes. Therefore, the inhibition of ROS production may suppress the phosphorylation of the NF-κB signaling pathway [33]. This ROS-modulating property of idebenone aligns with the observed suppression of proinflammatory cytokines, suggesting a potential link between ROS and NF-κB inhibition. Additionally, the ability of idebenone to preserve mitochondrial function may indirectly affect inflammation. Healthy mitochondria potentially function in cellular homeostasis, immune regulation, and inflammatory pathways [34]. In the present study, both in vivo and in vitro experiments confirmed that idebenone inhibited the expression of inflammatory cytokines and NOx induced by inflammatory diseases. Several in vitro studies have explored the effects of idebenone on inflammatory pathways. Given that the NF-κB pathway regulates the expression of pro-various inflammatory cytokines such as TNF-α and IL-1β, we examined the effect of idebenone on the NF-κB pathway in this study. Idebenone inhibits NF-κB activation, which suppresses the production of interleukin and TNF-α [35]. Similarly, idebenone treatment inhibited NF-κB nuclear translocation and the expression of inflammatory cytokines in the present study. Collectively, these findings suggest the anti-inflammatory potential of idebenone at the molecular level.

Collectively, this study confirmed that idebenone has therapeutic effects in mice models of CLP-induced sepsis and LPS-induced inflammation, including increased survival rates, tissue damage recovery, and reduced expression of inflammatory genes. Additionally, in vitro experiments showed that idebenone treatment ameliorated inflammatory responses in macrophages by suppressing any LPS-induced expression of inflammatory genes, inhibiting the activation of the NF-κB signaling pathway, and suppressing ROS production. Overall, these results suggest a novel role for idebenone as an anti-inflammatory agent in inflammatory diseases. However, an extensive understanding of the mechanisms and clinical implications of idebenone is necessary to facilitate clinical application in the treatment of inflammatory diseases. Therefore, comprehensive clinical trials are required to validate these findings in humans.

## Figures and Tables

**Figure 1 antioxidants-13-00151-f001:**
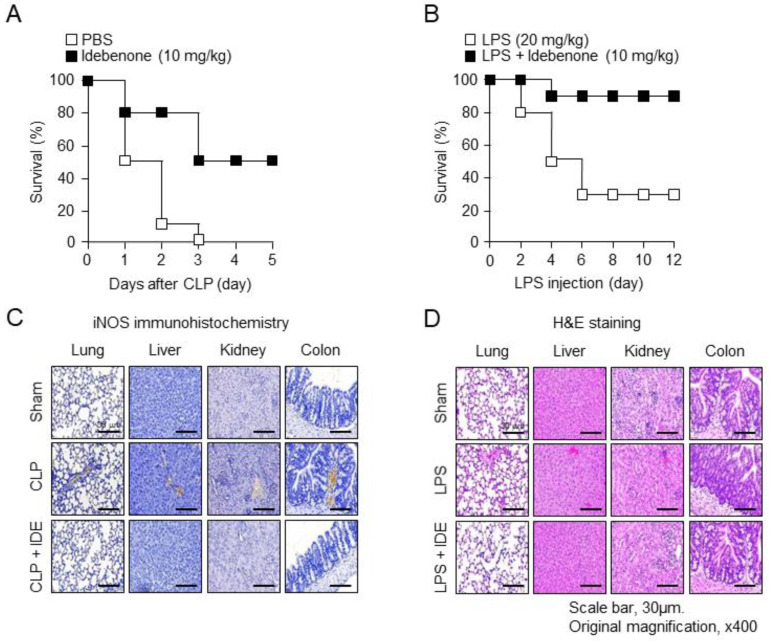
Idebenone prevents inflammation-induced mortality and inflammatory responses and restores tissue damage in vivo. Idebenone decreased the mortality rate of mice with CLP-induced sepsis and LPS-induced inflammation. (**A**) Mice were subjected to either CLP or sham operation (*n* = 10/group) and then intraperitoneally injected with idebenone (10 mg/kg). Mortality in the CLP + phosphate-buffered saline (PBS; sham operation, white) and CLP + idebenone (black) groups (*n* = 10/group) was monitored daily for five days after surgery. (**B**) C57BL/6 mice were intraperitoneally injected with LPS (20 mg/kg), followed by treatment with idebenone (10 mg/kg). Mortality in the LPS-only (white box) and LPS + idebenone (black box) groups (*n* = 10/group) was monitored daily for 12 days after injection. Idebenone protects against inflammation-induced tissue damaged in mice with CLP-induced sepsis and LPS-induced inflammation. (**C**) iNOS expression in the lung, liver, kidney, and colon tissues of mice in the sham and CLP + idebenone groups was examined using immunohistochemical analysis. (**D**) HE staining was performed to examine histological changes in the lung, liver, kidney, and colon tissues of mice in sham and LPS + idebenone groups. All tissues were collected in each group at 24 h after CLP surgery and LPS injection, with or without idebenone administration.

**Figure 2 antioxidants-13-00151-f002:**
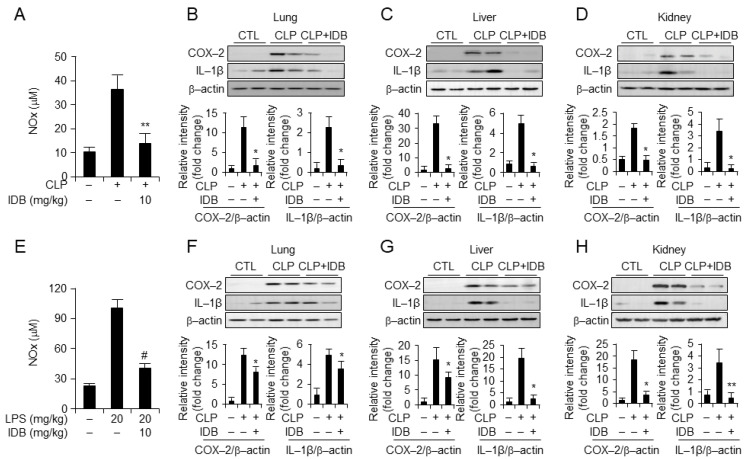
Idebenone inhibits CLP- and LPS-induced inflammatory responses. Idebenone inhibits NO metabolites (NO), COX-2, and IL-1β levels in murine models of CLP- and LPS-induced inflammatory models. The secretion of NO metabolites after CLP (**A**) and LPS (**E**) were determined using NO assays. Protein expression of COX-2 and IL-1β in the (**B**) lung, (**C**) liver, and (**D**) kidney tissues of mice in the sham, CLP, and CLP + idebenone groups was analyzed using Western blotting. Protein expression of COX-2 and IL-1β in the (**F**) lung, (**G**) liver, and (**H**) kidney tissues of mice in the sham, LPS only, and LPS + idebenone groups was analyzed using Western blotting. All tissues were randomly collected from two mice per group at 24 h after CLP and LPS; β-actin was used as a loading control; the quantification of the blots described in (**B**–**D**,**F**–**H**) was performed using ImageJ software. Statistical analyses were performed using paired two-tailed Student’s *t*-test. * *p* < 0.05, ** *p* < 0.01, # *p* < 0.001.

**Figure 3 antioxidants-13-00151-f003:**
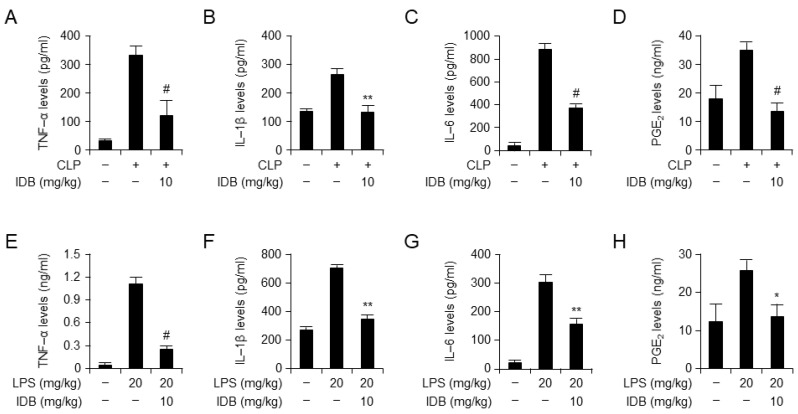
Idebenone inhibits the inflammatory response in mouse models of CLP- and LPS-induced inflammation. Idebenone inhibits TNF-α, IL-1β, IL-6, and PGE_2_ production in CLP- and LPS-induced inflammatory disease models. Serum levels of (**A**) TNF-α, (**B**) IL-1β, (**C**) IL-6, and (**D**) PGE_2_ following treatment with or without idebenone after CLP were determined using the ELISA assay. Serum levels of (**E**) TNF-α, (**F**) IL-1β, (**G**) IL-6, and (**H**) PGE_2_ following treatment with or without idebenone after the LPS injection were determined using the ELISA assay. *n* = 9/group; Graph represents the mean of three independent mouse serums levels of indicated mediators and cytokines. CLP-induced and LPS-induced mouse serum and tissue sample analyses were carried out by one-way ANOVA with Tukey’s Method using the GraphPad Prism 9 statistical program. Statistical significance was set at * *p* < 0.05, ** *p* < 0.01, and # *p* < 0.001.

**Figure 4 antioxidants-13-00151-f004:**
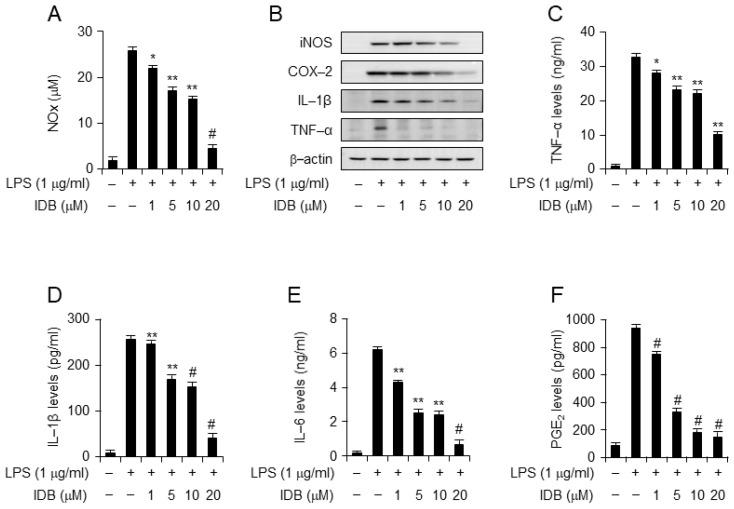
Idebenone inhibits LPS-induced expression of pro-inflammatory mediators in RAW 264.7 cells. Idebenone inhibited LPS-induced increase in the levels of NO metabolites, TNF-α, IL-1β, IL-6, and PGE_2_ in RAW 264.7 cells. (**A**) NO metabolite secretion was assessed using Griess reagent. (**B**) Western blotting was used to analyze the protein expression of inflammatory enzymes (iNOS, COX-2) and pro-inflammatory cytokines (IL-1β, TNF-α) in LPS-stimulated RAW 264.7 cells in the presence or absence of idebenone (1, 5, 10, 20 μM). Cells were pretreated with idebenone (1, 5, 10, 20 μM) for 2 h before LPS treatment (1 μg/mL). After 18 h, the cells were harvested for Western blotting, with β-actin as the loading control. The expression of (**C**) TNF-α, (**D**) IL-1β, (**E**) IL-6, and (**F**) PGE_2_ in LPS-stimulated RAW 264.7 cells in the presence or absence of idebenone was determined using the ELISA assay. Graphs depict the mean of three independent experiments. Statistical analysis was performed using paired two-tailed Student’s *t*-test. * *p* < 0.05, ** *p* < 0.01, # *p* < 0.001.

**Figure 5 antioxidants-13-00151-f005:**
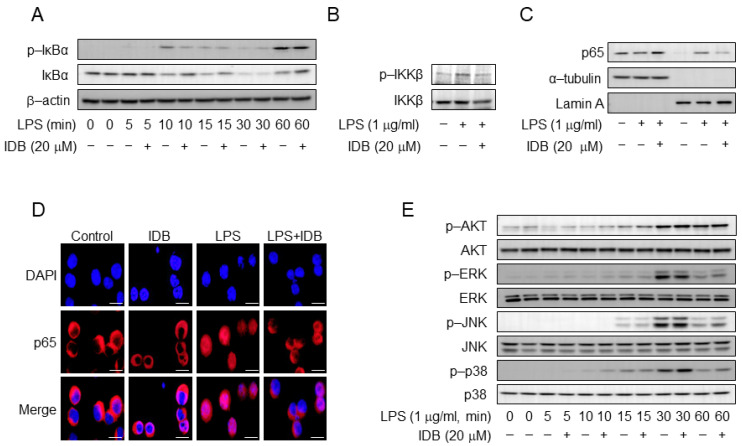
Idebenone inhibits the LPS-induced activation of NF-κB in macrophages. (**A**) The phosphorylation levels of IκBα and (**B**) IKKβ in RAW 264.7 cells were examined using Western blotting analysis. The cells were pre-incubated with or without idebenone (20 μM) for 2 h and subsequently treated with 1 μg/mL of LPS for the indicated times. (**C**) Western blot analysis for p65 expression in the cytosol and nuclear fraction of RAW 264.7 cells treated with or without idebenone (20 μM), followed by stimulation with 1 μg/mL of LPS for 30 min; Lamin A and tubulin were used as nuclear and cytosolic fraction loading controls. (**D**) The nuclear translocation of the NF-κB p65 subunit in RAW 264.7 cells pretreated with or without 20 μM of idebenone for 2 h, followed by stimulation with 1 μg/mL of LPS for 30 min, was detected using immunofluorescence staining. Scale bar = 50 μm. (**E**) The phosphorylation levels of AKT, ERK, JNK, and p38 in macrophages were examined using Western blotting analysis. Macrophages were pre-incubated with or without idebenone (20 μM) for 2 h and subsequently stimulated with 1 μg/mL of LPS for the indicated times; β-actin was used as a loading control.

**Figure 6 antioxidants-13-00151-f006:**
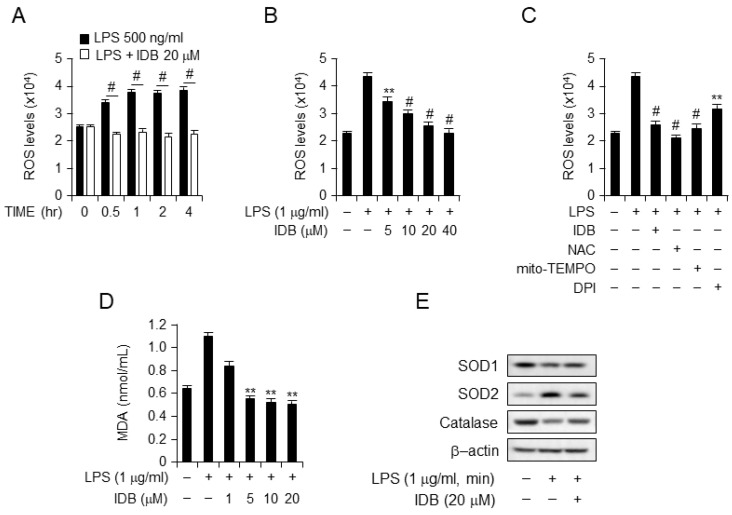
Idebenone suppresses ROS generation and lipid peroxidation in LPS-induced macrophage. (**A**) ROS production in DCFDA-stained RAW 264.7 cells was detected using flow cytometry. The cells were preincubated with or without 20 μM of idebenone for 2 h, followed by incubation with a medium containing 1 μg/mL of LPS for the indicated times. (**B**) The cells were preincubated with or without of idebenone (5, 10, 20, 40 μM) for 2 h, and then incubated with a medium containing 1 μg/mL of LPS for 2 h. (**C**) The cells were preincubated with 20 μM of idebenone and the ROS inhibitors NAC (5 mM), mito-TEMPO (50 μM), and DPI (10 μM) for 2 h, followed by incubation in a medium containing 1 μg/mL of LPS for 2 h. (**D**) MDA production in RAW 264.7 cells was measured using a lipid peroxidation assay kit. The cells were preincubated with idebenone (1, 5, 10, 20 μM) for 2 h, followed by incubation in a medium containing 1 μg/mL of LPS for 12 h. (**E**) RAW 264.7 cells were harvested, and the levels of SOD1 [Superoxide dismutase 1 (Cu-Zn)], SOD2 [Superoxide Dismutase 2 (MnSOD)], and catalase were measured using Western blotting. The cells were pretreated with 20 μM of idebenone for 2 h, followed by incubation in a medium containing 1 μg/mL of LPS for 12 h. Statistical analyses were performed using paired two-tailed Student’s *t*-test. ** *p* < 0.01, # *p* < 0.001.

## Data Availability

Data is contained within the article and Appendix A.

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
