# Peer review of "Anti-Inflammatory Effects of Idebenone Attenuate LPS-Induced Systemic Inflammatory Diseases by Suppressing NF-κB Activation"

_antioxidants, 2024, doi:10.3390/antiox13020151_

Round 1
Reviewer 1 Report
Comments and Suggestions for Authors
In the manuscript entitled "Anti-Inflammatory Effects of Idebenone Attenuate Systemic Inflammatory Diseases by Suppressing NF-κB Activation" by Choi et al., the authors found that idebenone, a synthetic analog of coenzyme Q10, improves survival rate and downregulates the expression of inflammation-related genes in model mice of cecal ligation puncture-induced sepsis and lipopolysaccharide-induced systemic inflammation. They also found that idebenone exerts anti-inflammatory activity by modulating the NF-κB signaling pathway. The study was well-designed and the results are clear. The comments of this reviewer are shown below.
Comments:
What kind of software did the authors use for statistical analysis? What is the p-value against? The authors cannot use the Student t-test because they compare more than two samples in one experiment. They should use a multiple comparison test such as the Tukey test.
In animal experiments, the authors found that the expression of inflammation-related genes is suppressed by idebenone in the lung, liver, and kidney; however, there is no data regarding macrophages collected from the model mice. So, why did the authors focus on the anti-inflammatory effect of idebenone on macrophages?
Although the authors conducted two animal studies, they show only one approval number. Is it correct?
In the Materials and Methods section, there is no description of how the lipopolysaccharide-induced systemic inflammation model mice were treated with idebenone.
In the Materials and Methods section, the authors mention that idebenone was prepared in phosphate-buffered saline for injection into mice; however, idebenone is insoluble in water. So, how did the authors prepare idebenone for administration to mice?
Lines 209-210: The authors stated that idebenone treatment increased the survival rate by approximately 80–90% at 6 days after LPS stimulation compared with that in the LPS-only group. Is this description correct?
Lines 349-350: The authors stated that idebenone treatment normalized the activities of antioxidant enzymes in LPS-stimulated macrophages. If they want to say so, they need to perform statistical analysis of that data.
Figure 1A: Show the number of mice per group.
Figures 2A and 2E: The authors showed the method for the measurement of levels of nitric oxide metabolites using the culture medium in the Materials and Methods section. So, which cells were collected and cultured from model mice of cecal ligation puncture-induced sepsis and lipopolysaccharide-induced systemic inflammation for measuring nitric oxide metabolites? There is no information in the manuscript.
Figure 3: Why were only three mice used for measurements?
Figure 5C: How were nuclear and cytoplasmic proteins prepared?
Author Response
Author’s Reply to the Review Report (Reviewer 1)
Please provide a point-by-point response to the reviewer’s comments and either enter it in the box below or upload it as a Word/PDF file. Please write down "Please see the attachment." in the box if you only upload an attachment.
We thank the Editor and the Reviewers for their detailed and helpful comments. We have revised the manuscript accordingly as described in the point-by-point response below. To this end, we have performed all the additional experiments suggested by the reviewers, resulting in the generation of additional figure in rebuttal letter and modified manuscript. We hope that that these changes, including the new experimental work, will render the manuscript suitable for publication in Antioxidant Journal.
We have extensively addressed all the other concerns of the Reviewers as detailed below.
In the manuscript entitled "Anti-Inflammatory Effects of Idebenone Attenuate Systemic Inflammatory Diseases by Suppressing NF-κB Activation" by Choi et al., the authors found that idebenone, a synthetic analog of coenzyme Q10, improves survival rate and downregulates the expression of inflammation-related genes in model mice of cecal ligation puncture-induced sepsis and lipopolysaccharide-induced systemic inflammation. They also found that idebenone exerts anti-inflammatory activity by modulating the NF-κB signaling pathway. The study was well-designed and the results are clear. The comments of this reviewer are shown below.
Comments:
What kind of software did the authors use for statistical analysis? What is the p-value against? The authors cannot use the Student t-test because they compare more than two samples in one experiment. They should use a multiple comparison test such as the Tukey test.
We thank the Reviewer for the strong supportive comments. We conducted experiments using the one-way ANOVA statistical method with Tukey's Method for mouse serum samples. We have described the experimental procedure and results of this study in the 'Materials and Methods' section and represented them in the figure legends of the manuscript.
“CLP-induced and LPS-induced mouse serum and tissue samples analyses were carried out by one-way ANOVA with Tukey’s Method using the GraphPad Prism 9 statistical program. Statistical significance was set at *p < 0.05, **p < 0.01, and #p < 0.001.”
In animal experiments, the authors found that the expression of inflammation-related genes is suppressed by idebenone in the lung, liver, and kidney; however, there is no data regarding macrophages collected from the model mice. So, why did the authors focus on the anti-inflammatory effect of idebenone on macrophages?
We thank the Reviewer for the strong supportive comments.
Although we did not mention it in the manuscript, we analyzed the anti-inflammatory effects of idebenone after inducing LPS using bone marrow-derived macrophages (BMDMs) differentiated from mouse bone marrow. The experimental results of idebenone's anti-inflammatory effects in BMDMs after LPS induction showed that NOx exhibited statistically significant anti-inflammatory effects 24 hours later, depending on the dose. However, for pro-inflammatory cytokines such as TNF-a, IL-1b, and IL-6, there was only an effect at an idebenone concentration of 20 μM. These results indicated that while macrophage cell lines like RAW264.7 or J774A.1 cells showed statistically significant effects at concentrations of 5-10 μM, BMDMs exhibited relatively lower cytokine inhibition effects. As a result, we did not include the ex vivo results in the manuscript. Nevertheless, we concluded that macrophages could play an important role in various tissues and proceeded with subsequent studies. We kindly ask the reviewers for their broad understanding.
Although the authors conducted two animal studies, they show only one approval number. Is it correct?
We thank the Reviewer for the strong supportive comments.
Our approval number encompasses comprehensive animal experimental methods for mouse inflammation experiments related to sepsis, allowing for two different animal experiments under a single approval number.
In the Materials and Methods section, there is no description of how the lipopolysaccharide-induced systemic inflammation model mice were treated with idebenone.
We thank the Reviewer for the strong supportive comments. We added the following content on lipopolysaccharide-induced systemic inflammation mouse model experimental method (highlighted part) in the Materials and Methods section, according to the reviewer's comments
“Systemic inflammation was induced by intraperitoneal injection of lipopolysaccharide (LPS) at a dose of 10 mg/kg in a volume of 100 μL sterile saline in the experimental groups. To obtain 200 µg of idebenone from the 40 mM idebenone stock solution in DMSO, approximately 0.6 µL of the idebenone stock solution is needed. This was then diluted with PBS to a total volume of 1 mL to overcome the insolubility of idebenone and for fluid resuscitation. Idebenone (dissolved in DMSO, 10 mg/kg) was administered intraperitoneally to the treatment group immediately following LPS injection. A control group received intraperitoneal injections of the vehicle. After injection, mice were monitored for 12 days post-treatment, and blood, tissues, or relevant samples were collected for subsequent analyses.”
In the Materials and Methods section, the authors mention that idebenone was prepared in phosphate-buffered saline for injection into mice; however, idebenone is insoluble in water. So, how did the authors prepare idebenone for administration to mice?
We selected DMSO as the solvent considering the insoluble activity of 25 mg idebenone (Cayman cat NO: 15475), and prepared a stock solution at 40 mM. Subsequently, after inducing inflammation in 8-week-old mice with LPS, we proceeded with an intraperitoneal (ip) injection of idebenone at 10 mg/kg to analyze its anti-inflammatory efficacy. By using this method, we were able to overcome the issue of idebenone's insolubility in water.
<Actual amount of idebenone needed for injection>
Idebenone (10,000 µg) / body weight (1,000 g) = required amount of idebenone (X µg)/mouse (~20 g)
Required amount of idebenone (X) = 200 µg, To obtain 200 µg of idebenone from the 40 mM idebenone stock solution in DMSO, approximately 0.6 µl of the idebenone stock solution is needed. This was then diluted with PBS to a total volume of 1 ml and injected intraperitoneally into the mice.
Lines 209-210: The authors stated that idebenone treatment increased the survival rate by approximately 80–90% at 6 days after LPS stimulation compared with that in the LPS-only group. Is this description correct?
We thank the Reviewer for the strong supportive comments. Based on the reviewer's comments, we have restructured the mentioned sentence as follows (line 209-210 highlighted part) and incorporated it into the result section of the manuscript with modifications for English grammar and flow.
“After intraperitoneal injection of idebenone (10 mg/kg), the group compared to the PBS group following CLP surgery showed an improvement in survival rate of approximately 40-50% by the 3rd day. (Figure 1A). Similarly, the LPS-only group exhibited an approximately 80% survival rate on the 2nd day after LPS injection, which decreased to approximately 50% on the 4th day, and was maintained at 30% on the 6th day. In contrast, the LPS plus idebenone group consistently maintained an 80% survival rate starting from the 4th day after LPS injection.”
Lines 349-350: The authors stated that idebenone treatment normalized the activities of antioxidant enzymes in LPS-stimulated macrophages. If they want to say so, they need to perform statistical analysis of that data.
We thank the Reviewer for the strong supportive comments. Based on the reviewer's comments, we have restructured the mentioned sentence as follows (line 349-350 highlighted part) and incorporated it into the result section of the manuscript with modifications for English grammar and flow.
“Western blotting revealed that Idebenone treatment reinstates the expression of antioxidant enzymes, including SOD1, SOD2, and catalase, in LPS-stimulated macrophages, effectively restoring their levels to normal. (Figure 6E)”
Figure 1A: Show the number of mice per group.
We apologize for the oversight in properly labeling the mice in each group. In response to this, we have addressed the issue by inserting the notation indicating the number of mice per group into the legend of Figure 1A, and have highlighted this information accordingly.
“(A) Mice were subjected to either CLP or sham operation (n = 10/group) and then intraperitoneally injected with idebenone (10 mg/kg). Mortality in the CLP + phosphate-buffered saline (PBS; sham operation, white) and CLP + idebenone (black) groups (n = 10/group) was monitored daily for five days after surgery.”
Figures 2A and 2E: The authors showed the method for the measurement of levels of nitric oxide metabolites using the culture medium in the Materials and Methods section. So, which cells were collected and cultured from model mice of cecal ligation puncture-induced sepsis and lipopolysaccharide-induced systemic inflammation for measuring nitric oxide metabolites? There is no information in the manuscript.
After carefully reviewing the reviewer's comments, we have made additional revisions to the manuscript to address the insufficient explanations. Figures 2A and 2E represent the levels of nitrite and nitrate (NOx) measured in the serum. In order to differentiate these findings from the cytokine expression observed in the tissues, we have restructured the sentences accordingly. Additionally, to provide further details on the experimental method, we have included information about the nitrate reductase-based colorimetric assay kit used in the experiments in the Materials and Methods section. The following content has been added to the manuscript.
<Result Section> “Idebenone treatment demonstrated inhibition of NOx increase in the serum of the CLP-induced mouse model (Figure 2A), as well as a significant suppressive effect on the upregulation of inflammatory cytokines, including COX-2 and IL-1beta, in lung, liver, and kidney tissues (Figure 2B-D). Moreover, idebenone treatment effectively attenuated the LPS-induced inflammatory response, as indicated by reduced serum NOx levels (Figure 2E) and suppressed expression of inflammatory cytokines in lung, liver, and kidney tissues (Figure 2F-H)”.
<Material and Method> Serum nitrite plus nitrate (NOx) concentration was determined with a nitrate reductase-based colorimetric assay kit (Alexis, CA, USA).
Figure 3: Why were only three mice used for measurements?
I sincerely apologize for not carefully reviewing the manuscript before submission. In reality, we conducted the experiment three times, with three mice in each repetition, resulting in a total of nine mice used in the study. However, there seems to have been an oversight where it was mistakenly recorded as three mice in the documentation. We have made the necessary correction in the manuscript to accurately reflect that a total of nine mice were used in the study.
I kindly request the reviewer's understanding and generosity regarding this mistake. Thank you very much.
Figure 5C: How were nuclear and cytoplasmic proteins prepared?
We thank the Reviewer for the strong supportive comments. According to reviewer’s comments, we provided “Cytoplasmic and nuclear fractionation methods” in the material and method section as below.
“Preparation of cytoplasmic and nuclear fractions
After treating RAW264.7 cells with lipopolysaccharide (LPS, 1 μg/mL) in the presence or absence of idebenone (20 μM) for 30 minutes, cellular nuclear and cytoplasmic fractions were isolated using a commercial kit (Active Motif, Carlsbad, USA) following the manufacturer's instructions. The fractions were obtained to study the translocation of NF-κB, a crucial transcription factor in the inflammatory response. To ensure consistent protein loading for subsequent Western blot analysis, the concentration of each fraction was determined using a Pierce BCA assay (Thermo Scientific). Fractionation efficiency was assessed by Western blotting using α-tubulin as a loading control for the cytoplasmic fraction and Lamin A as a loading control for the nuclear fraction.”

Reviewer 2 Report
Comments and Suggestions for Authors
Choi et al. demonstrated that idebenone ameliorated CLP- and LPS-induced systemic sepsis in mice model and inhibited LPS-stimulated activation of NF-κB signaling in two murine macrophage cell lines. This study was fundamentally sound science and the manuscript is entirely well organized. However, it seems that the paper cannot be published unless flaws in the data analysis using western blotting and errors in the statistical analysis are corrected. There are also many minor points. Please improve the individual points below.
Major points:
1. Title and Results section: “Systemic Inflammatory Diseases” encompass a wide range of diseases, including not only infectious diseases but also metabolic diseases such as obesity and diabetes. Regarding the title and results, therefore, it is more accurate to describe it as “CLP- and LPS-induced systemic sepsis”. However, it is possible to speculate that idebenone is effective against systemic inflammatory diseases.
2. T-tests cannot be used when comparing three or more groups. After applying one-way ANOVA, if a significant difference is found, use a post-hoc test to evaluate whether there is a significant difference between each group. As a post-hoc test, Dunnett’s test should be used to analyze significant differences between each test group and the control group, while Tukey’s test, etc. should be used to analyze those between all groups. There should be some free statistical analysis software.
3. Bands detected by western blotting typically need to be quantified using ImageJ free software and subjected to statistical analysis. It is not appropriate to conclude an increase or decrease based on one or two pieces of data. However, regarding Figure 5A and 5E, it may be considered enough to analyze only representative times.
Minor points:
1. Line 92: "Under standard conditions" should include specific room temperature and humidity.
2. Line 103: By which route was 1 mL of idebenone or vehicle injected?
3. Line 122: “by washing three times with --- (5 min each)”
4. Line 126: What are the ingredients of the “blocking solution”?
5. Line 128, 130 and 132: Is the “buffer” Tris-based or phosphate-based?
6. Line 136: “icroscope.”?
7. Line 140: “, was” should be deleted.
8. Line 141 and 144: Are “calf serum” and “fetal bovine serum” different? Are they heat inactivated?
9. Line 148: “levels NO” should be “levels of NO”.
10. Line 156: Is “phosphatase inhibitor cocktail” also supplemented in addition to the “protease inhibitor cocktail”?
11. Line 165: What is used as the exposure device?
12. Line 210: “challenge” or “injection” may be better than “stimulation”.
13. Line 242: “E-G” should be “E-H”.
14. Line 234-235 and 256: These subtitles need to be differentiated or integrated.
15. Line 292: “levels NO” should be “levels of NO”.
16: Line 327: Cytoplasmic and nuclear fractionation methods should be described in the “Materials and Methods” section.
17: Line 349 and 393: “activities (or activity)” should be “expressions (or expression)” because enzymatic activity was not analyzed in this study.
18. Line 407: “anti-” is “pro-”?
19. Line 410: “NF-κB phosphorylation” should be “NF-κB nuclear translocation”?
Comments on the Quality of English LanguagePlease see the Comments and Suggestion for Authors.
Author Response
Author’s Reply to the Review Report (Reviewer 2)
Choi et al. demonstrated that idebenone ameliorated CLP- and LPS-induced systemic sepsis in mice model and inhibited LPS-stimulated activation of NF-κB signaling in two murine macrophage cell lines. This study was fundamentally sound science and the manuscript is entirely well organized. However, it seems that the paper cannot be published unless flaws in the data analysis using western blotting and errors in the statistical analysis are corrected. There are also many minor points. Please improve the individual points below.
Major points:
- Title and Results section: “Systemic Inflammatory Diseases” encompass a wide range of diseases, including not only infectious diseases but also metabolic diseases such as obesity and diabetes. Regarding the title and results, therefore, it is more accurate to describe it as “CLP- and LPS-induced systemic sepsis”. However, it is possible to speculate that idebenone is effective against systemic inflammatory diseases.
We thank the Reviewer for the strong supportive comments.
We carefully considered the reviewer's comments and, with the intention of accepting them, decided to modify the title as follows.
“Anti-Inflammatory Effects of Idebenone Attenuate LPS-induced Systemic Inflammatory Diseases by Suppressing NF-κB Activation.”
- T-tests cannot be used when comparing three or more groups. After applying one-way ANOVA, if a significant difference is found, use a post-hoc test to evaluate whether there is a significant difference between each group. As a post-hoc test, Dunnett’s test should be used to analyze significant differences between each test group and the control group, while Tukey’s test, etc. should be used to analyze those between all groups. There should be some free statistical analysis software.
We thank the Reviewer for the strong supportive comments. The question above can be considered almost identical to the third question from reviewer 1. We conducted experiments using the one-way ANOVA statistical method with Tukey's Method for mouse serum samples. We have described the experimental procedure and results of this study in the 'Materials and Methods' section and represented them in the figure legends of the manuscript.
“CLP-induced and LPS-induced mouse serum and tissue samples analyses were carried out by one-way ANOVA with Tukey’s Method using the GraphPad Prism 9 statistical program. Statistical significance was set at *p < 0.05, **p < 0.01, and #p < 0.001.”
- Bands detected by western blotting typically need to be quantified using ImageJ free software and subjected to statistical analysis. It is not appropriate to conclude an increase or decrease based on one or two pieces of data. However, regarding Figure 5A and 5E, it may be considered enough to analyze only representative times.
We thank the Reviewer for the strong supportive comments. After quantifying the band detected by our western blot using ImageJ free software, statistical analysis was performed to add the graph to Fig.2. And we added the image of the additional western blot to the western blot raw data file.
“Quantification of the western blots were analyzed Image J software, and the level of the indicated protein was normalized to β-actin. Significant differences between groups were determined using two-tailed unpaired Student’s t-test.”
Minor points:
- Line 92: "Under standard conditions" should include specific room temperature and humidity.
We thank the Reviewer for the strong supportive comments.
We added up statement for the animal facility standard condition in line 106-107 (highlighted part which is animal experimentations) as below, instead of “under standard conditions”.
“housed in plastic cages in a room with controlled temperature (22 1 ± 2°C) and humidity (55 ± 5%) and maintained on a reverse 12h light/dark cycle.”
- Line 103: By which route was 1 mL of idebenone or vehicle injected?
We thank the Reviewer for the helpful comments. We added up the statement injection route in the line 116-119 (highlighted part) and replaced the sentence as below.
“At 24 h after the sham and CLP operations, the mice were either injected with 1 mL of idebenone (10 mg/kg) prepared in phosphate buffered saline (PBS) for anti-inflammatory drug effect and fluid resuscitation or with 1 mL of PBS only subcutaneously for fluid resuscitation.”
- Line 122: “by washing three times with --- (5 min each)”
We apologize for the lack of clarity. We added up the statement which is the buffer name “xylene” in the line 137-138 as you mentioned “by washing three times with---“ and also replaced the sentence in the material and methods as below.
“the sections were deparaffinized by washing three times with xylene (5 min each),”
- Line 126: What are the ingredients of the “blocking solution”?
According to the reviewer's comment, we have added information about the composition of the blocking solution to the manuscript as follows:
“blocking solution with 1.5% bovine serum albumin (BSA) in Tris-buffered saline (TBS; 50 mM Tris and 150 mM NaCl, pH 7.6) containing 0.1% Tween 20 (TBS-T) for 1 h at room temperature (25°C),”
- Line 128, 130 and 132: Is the “buffer” Tris-based or phosphate-based?
According to the reviewer's comment, we have revised the ambiguous expression, grammar errors, and enhanced the natural flow of the manuscript. Additionally, to improve clarity, we have replaced 'wash buffer' with 'TBS-T' throughout the manuscript.
“Thereafter, the samples were washed three times with TBS-T for 5 minutes, followed by incubation with 200 µL of secondary antibody for 30 minutes at room temperature. After washing three times with TBS-T for 5 minutes, the samples were treated with ABC reagent kit (Mouse IgG PK-6102, Vector Laboratories, Burlingame, CA). Thereafter, the sections were washed three times with TBS-T and stained with 100 µL of DAB, rinsed twice with dH2O for 5 minutes, treated twice with 95% ethanol for 10 seconds, and then twice with 100% ethanol.”
- Line 136: “icroscope.”?
According to reviewer’s comment, we fixed wrong spelling microscope. icroscope was replaced with “microscope” in the line 151 (highlighted part (material and methods “Immunohistochemistry for iNOS”).
- Line 140: “, was” should be deleted.
We thank the Reviewer for the careful comments. We deleted it which is “was” in the line 156 (highlighted part (material and methods “cells and cell culture”).
- Line 141 and 144: Are “calf serum” and “fetal bovine serum” different? Are they heat inactivated?
We thank the Reviewer for the generous comment. For clarifying the lack of material and methods, following sentences were added up revised manuscripts in line 156-162 as below.
“RAW 264.7 cells were cultured in Dulbecco’s modified Eagle’s medium (SH30243.01, Hyclone, UT, USA) supplemented with 10% non-heat inactivated calf serum (26170-043, GibcoTM, MA, USA) and 1% antibiotics (LS 203-01, WELGENE Inc., Gyeongsan, Korea). The J774A.1 cell line was cultured in Dulbecco’s modified Eagle’s medium (SH30243.01, Hyclone, UT, USA) supplemented with 10% non-heat inactivated fetal bovine serum (SH30919.03, Hyclone, UT, USA), 1% antibiotics (LS 203-01, WELGENE Inc., Gyeongsan, Korea), and…..” (line 156-162)
- Line 148: “levels NO” should be “levels of NO”.
We thank the Reviewer for the supportive comment. We replaced the title “Measurement of levels NO metabolites” to “Assessment of NO metabolite levels” in line 165 (highlighted part)
- Line 156: Is “phosphatase inhibitor cocktail” also supplemented in addition to the “protease inhibitor cocktail”?
We thank the Reviewer for the supportive comment.
Our lab’s RIPA buffer includes components such as sodium fluoride, sodium orthovanadate, b-glycerophosphate, and sodium pyrophosphate, all of which possess the ability to inhibit phosphatases. For this reason, I attempted treatment with a protease inhibitor cocktail after adapting the RIPA buffer. Additionally, I will also include the RIPA buffer recipe in the revised manuscript in line 173-176 (highlighted part) as below.
“(RIPA) lysis buffer (50 mM Tris-HCl, pH 7.6, 150 mM NaCl, 10% glycerol, 1% Nonidet P-40, 5 mM EDTA, 1 mM DTT, 100 mM NaF, 2 mM sodium pyrophosphate, 20 mM b-glycerophophate, 2 mM sodium orthovanadate)”
- Line 165: What is used as the exposure device?
We apologize for the lack of clarity. We have incorporated details regarding the exposure device utilized for capturing chemiluminescent images by Fusion Solo S (Vilber Lourmat, Paris, France) into the revised manuscript, addressing the need for comprehensive technical information. Please find the updated section in the manuscript line 183-186 (highlighted part) as below.
“chemiluminescent images were captured by Fusion Solo S (Vilber Lourmat, Paris, France) following the instruction manual”
- Line 210: “challenge” or “injection” may be better than “stimulation”.
We thank the Reviewer for the helpful suggestions. According to Reviewer’s comment, we have replaced the mentioned sentence as follows (line 210 highlighted part) and incorporated it into the result section of the manuscript with modifications for English grammar and flow.
“After intraperitoneal injection of idebenone (10 mg/kg), the group compared to the PBS group following CLP surgery showed an improvement in survival rate of approximately 40-50% by the 3rd day. (Figure 1A). Similarly, the LPS-only group exhibited an approximately 80% survival rate on the 2nd day after LPS injection, which decreased to approximately 50% on the 4th day, and was maintained at 30% on the 6th day. In contrast, the LPS plus idebenone group consistently maintained an 80% survival rate starting from the 4th day after LPS injection.”
- Line 242: “E-G” should be “E-H”.
We apologize for the lack of clarity. According to reviewer’s comment, we fixed wrong labeling from “E-G” to “E-H” in the line 251 of the results section (highlighted part).
- Line 234-235 and 256: These subtitles need to be differentiated or integrated.
We agreed with reviewer’s comment and after that we integrated with these subtitles. We deleted the line 256 which is post subtitle and combine figure 2 paragraph (results part) with figure 3 paragraph (result part).
- Line 292: “levels NO” should be “levels of NO”.
We apologize for the lack of clarity. According to reviewer’s comment, we fixed wrong labeling from “levels NO” to “levels of NO” in the line 564 of the figure 4 legend section (highlighted part).
16: Line 327: Cytoplasmic and nuclear fractionation methods should be described in the “Materials and Methods” section.
We thank the Reviewer for the strong supportive comments. According to reviewer’s comments, we provided “Cytoplasmic and nuclear fractionation methods” in the material and method section as below.
“Preparation of cytoplasmic and nuclear fractions
After treating RAW264.7 cells with lipopolysaccharide (LPS, 1 μg/mL) in the presence or absence of idebenone (20 μM) for 30 minutes, cellular nuclear and cytoplasmic fractions were isolated using a commercial kit (Active Motif, Carlsbad, USA) following the manufacturer's instructions. The fractions were obtained to study the translocation of NF-κB, a crucial transcription factor in the inflammatory response. To ensure consistent protein loading for subsequent Western blot analysis, the concentration of each fraction was determined using a Pierce BCA assay (Thermo Scientific). Fractionation efficiency was assessed by Western blotting using α-tubulin as a loading control for the cytoplasmic fraction and Lamin A as a loading control for the nuclear fraction.”
17: Line 349 and 393: “activities (or activity)” should be “expressions (or expression)” because enzymatic activity was not analyzed in this study.
We thank the Reviewer for the generous comments. We substituted the term 'expression(s)' for 'activities' in lines 349 and 393. Additionally, we highlighted the term 'expression(s)' in the revised manuscript of results and discussion section after the substitution.
- Line 407: “anti-” is “pro-”?
We thank the Reviewer for the critical comments. Most of NF-kB transcription factor could regulate the expression of pro-inflammatory cytokines such as TNF-a and IL-1b in this manuscript. For this reason, we substituted the term 'pro- for 'anti- in lines 407.
“Given that the NF-κB pathway regulates the expression of various pro-inflammatory cytokines such as TNF-a and IL-1b, we examined the effect of idebenone on the NF-κB pathway in this study.”
- Line 410: “NF-κB phosphorylation” should be “NF-κB nuclear translocation”?
We sincerely appreciate the valuable feedback from the reviewer. In response to the reviewer's insightful comments, we have replaced the term 'NF-κB phosphorylation' with 'NF-κB nuclear translocation' in the line 410 (highlighted part).

Round 2
Reviewer 1 Report
Comments and Suggestions for Authors
The authors have addressed all of this reviewer's comments.
Reviewer 2 Report
Comments and Suggestions for Authors
The authors have significantly improved the paper. This paper is ready for publication.